# From protection of sacrificial self to critical turning points and growth: Redeployed nurses' experiences on the frontline during the COVID-19 pandemic

Sinéad Creedon[1¤a*], Anna Trace[2¤b]

1 School of Public Health, University College Cork, Cork, Ireland, 2 School of Applied Psychology, University College Cork, Cork, Ireland

¤a Current address: School of Public Health, Western Gateway Building, University College Cork, Cork, Ireland
¤b Current address: School of Applied Psychology, Cork Enterprise Centre, North Mall Campus, University College Cork, Cork, Ireland
* screedon@ucc.ie

## Abstract

The mental health and wellbeing of nurses has been a concern, long before the arrival of the COVID-19 pandemic. Working on the frontline under such challenging circumstances, for extended periods of time, has resulted in negative psychological responses. The current study aims to examine redeployed nurses' resilience in acute hospitals in Ireland, during a period of adversity (pandemic). The impact on their personal and professional identity, and their perception of meaningful supports and coping will be explored. An interpretative phenomenological analysis was carried out to gain insight into how nurses adapted to the changing work environment during the pandemic. Online semi-structured interviews were carried out with six experienced female nurses, who were redeployed to the frontline from their own roles. Three experiential themes representing the nurses' journey were generated: Protection of Sacrificial Self; The Fortifying Effect of Us, and Critical Turning Points & Growth. Nurses made significant sacrifices and had to find ways to detach to cope. They revitalized themselves by creating a sense of 'us' to help them face a harsh climate against others, which enabled critical turning points and growth. This study has strongly highlighted the emotional effects on nurses due to feeling isolated, undervalued, and excluded during redeployment to the frontline. It has also featured how well redeployed nurses coped while faced with an existential crisis. It has given voice to all nurses who faced this pandemic, despite exposure to a risk of burnout and threats to their mental health and wellbeing. This study has further enriched our understanding of personal growth and trauma in adverse work conditions by including an exploration of what sacrificial commitment adds to our understanding of physical and moral courage. Future provision of supports for nurses must be ongoing both during and after crisis events.

**Data availability statement:** De-identified extracts from qualitative transcripts are available only upon request as the participants consented to "extracts" from their interviews being quoted in the thesis, and in any subsequent publications, and not the full transcripts. Raw format (verbatim transcripts) contains both potentially identifying and sensitive participant information (e.g., detailed discussions, mental health challenges, potentially identifiable patient details, and participant geographical locations). Hence, there is a risk that sharing publicly would compromise participant anonymity and confidentiality. Data requests should be fielded to the School of Applied Psychology Research Ethics Committee (APREC), at University College Cork, at ethics.ap@ucc.ie Cleaned data (semi-structured interview guide, IPA 7-steps guide, copy of the consolidated criteria for reporting qualitative research (COREQ), copy of critical appraisal skills programme (CASP) checklist, can be accessed at https://doi.org/10.5281/zenodo.15603364 for this research article.

**Funding:** The author(s) received no specific funding for this work.

**Competing interests:** The authors have declared that no competing interests exist.

## Introduction

Long before the COVID-19 pandemic, the mental health and wellbeing of nurses has been a concern, due to the nature of their work, and also due to lack of resources during the last financial crash [1,2].

Redeployed nurses were "the hardest hit" [3]. Their workloads scaled up significantly both at work and at home. Female nurses propped up our healthcare system while also largely holding down the family home. This caused them much distress, overwhelm, burnout, and in some cases, even suicide [4].

Nurse 'redeployment' during the pandemic was when nurses were assigned without choice, from their own clinical settings to unfamiliar COVID wards on the frontline [5]. As a result, they felt forced into complex situations where they had to rapidly adjust, due to unprecedented workplace disruption. This resulted in them experiencing poorer physical and mental health outcomes [6]. It is important to note that not all nurses were redeployed to the frontline. These nurses were less affected than their redeployed colleagues as they did not have as many pandemic-related disruptions within their own clinical settings.

Women make up more than 85% of the global nursing workforce [7]. It is well documented that the pandemic exacerbated gender inequalities in women, which has introduced brand new challenges that are unprecedented to society at large [8]. As a result of the profound effect on a majority female workforce, this study aims to examine redeployed female nurse resilience during the COVID-19 pandemic in acute hospitals in Ireland. The impact on their personal and professional identity, and their perception of meaningful supports and coping will be explored.

Some of the key burdens facing redeployed nurses included the existential threat of death and dying, fear of contracting or spreading disease, moral injury, and pressure from being under public scrutiny. The existential threat of death and dying created a heightened state of alertness and resulted in sustained levels of hypervigilance [9,10]. Whereas feelings of fear of being infected, or infecting others, impacted their capacity to do their job, and impacted their emotional, psychological and physical wellbeing [11,12]. Moral injury was experienced by many frontline nurses. This is when there is prolonged exposure to traumatic events or feeling ill-prepared for adverse events leading to mental health challenges as well as compassion fatigue [13]. Compassion fatigue is the result of a cumulative process caused by intense, continuous, and prolonged contact with patients, while also being exposed to ongoing stress [14]. Additionally, the public portrayal of nurses as superheroes was a further burden, placing too much pressure and emphasis on nurses to deliver care at an unachievably high level [15].

Because this extra burden caused much distress and threat to nurses, they relied on potent war metaphors to help portray their trauma [10]. War metaphor themes coming from some of the research included 'hard armour', 'temporary concentration camps', and where nurses reported having 'won this battle' [10,16].

Despite being redeployed to the frontline, nurses' profound sense of duty and professional responsibility enabled them to stay, commit, and sacrifice themselves to a harsh climate, flanked by their peers [17].

In facing this adversity, nurses needed to rapidly build resilience [18]. Resilience is described as the ability to adapt well in the face of adversity while also maintaining normal psychological and physical functioning [19]. Research conducted on previous wars, natural disasters, and pandemics, confirm how important resilience is for nurses to be able to adapt and cope in stressful work situations [20]. Key strategies utilised by nurses to build resilience were, being emotionally tough [21] and being able to emotionally detach from work [22].

Furthermore, nurses' resilience was reduced during the pandemic due to feeling undervalued at work, being emotionally and physical exhausted, having poor organisational and social supports, unfairness or inequality at work, and the inability to care for patients to acceptable standards [15,23]. Interestingly, resilience training has long been contested by nurses as they often feel "it is another stick to beat them with" [15]. In other words, if nurses are seen to be struggling at work, whether due to moral injury, compassion fatigue or burnout, they sometimes feel it is their "fault" because they are not being "resilient enough" [15]. This can leave them feeling personally responsible for organisational failures which can add to an already burdensome workplace [24].

Nurses' self-coping mechanisms, social, and workplace supports, have been shown to strengthen resilience when faced with adversity at work. The importance of self-care has been shown to impact positively on frontline nurses [25,26]. Other research has found that detaching from social media, news, and government briefings was protective during the COVID-19 pandemic [15]. Some qualitative studies have shown that self-coping mechanisms such as introspection, engaging with nature, and spending time with family helped nurses to get through this crisis [27]. Notably, at times of severe adversity, other studies found a strong link between spirituality and improved health, such as self-acceptance, and the ability to forgive self [28].

It was found that positioning trained and experienced psychologically savvy clinical nurse managers on the frontline, was another way to support redeployed nurses [29]. For example, Greenberg and colleagues strongly recommend that nurse managers needed to be on watch for nurses who may be struggling and only offer 'psychological PPE' (not debriefing) by adopting a 'nipping it in the bud' approach. This provided at-risk nurses with a focus on early 'return to duty' where they were actively monitored in a 'watch and wait approach' [30]. Finally, providing nurses with a clear pathway to access professional help, engaging in trauma risk management, debriefing and peer support at a later stage if required, was recommended [31].

Frontline nurses were touched to see some of their senior nurse managers step up. Managers did this by being physically present whilst acknowledging challenges, even if they didn't have all the solutions [32]. Another study conducted in the UK identified how little physical presence there was of senior nurse managers on the frontline as some of them prioritised policy and procedure over compassionate leadership and emotional support for their nursing teams [18].

There were also times when supports were less helpful. Although nurses were brought closer by their emotionally shared experiences, relationships with their peers became strained, especially where workloads were reported to be unequal [25]. Another example was where nurses felt valued by the support they received from the public. This boosted morale initially but was later described by nurses as a 'double edged sword' where most felt that support was short-lived. As a result, they became cynical of their relationship with the public, once that support waned [11,12,25]. Nurses who sought support from their families often found it difficult to share their experiences, as they wanted to protect their loved ones from the harsh realities behind hospital doors. Although everyone was facing the same crisis, the nurses felt they were not 'in the same boat' as their families. As a result, they did not receive the same level of support from their families as they did from their fellow nurse colleagues [33].

In Ireland, it is important to highlight that despite the Health Service Executive's (HSE) significant investment in organisational mentoring and coaching, there was no formal attempt to roll out supports through these existing services, either during or since the pandemic. This may have been beneficial in supporting redeployed nurses, while also facilitating their post-traumatic growth and adaptation. Post-traumatic growth is the positive psychological change that occurs as a result of struggling with highly challenging circumstances [34].

For redeployed nurses, those with higher levels of perceived solidarity with social groups, were found to have a stronger sense of purpose and pride in their work, resulting in growth [4]. Nurses can experience post traumatic growth after traumatic events. This higher level of functioning can coexist with the ongoing distress experienced during traumatic events [35]. Because nurses are perceived to be caring and compassionate, practicing those virtues and knowing their own personal strengths can help bring some meaning to their adversity and traumatic experiences. This also enables them to build hope, self-efficacy, and to reinstate some work life balance [16].

To conclude, many studies have taken place in various countries exploring healthcare worker resilience, and the challenges they faced during the pandemic [16]. Other studies focused specifically on nurses' experience on the frontline [32]. However, further understanding and insight into Irish nurse's experiences on the frontline is needed, to understand more about their resilience, and what was supportive for them. This is particularly important given how stretched Irish healthcare workers have been for so long, due to austerity, attrition rates, and extended crisis events such as the more recent COVID-19 pandemic.

## Materials and methods

### Aims

The aim of this study was to examine redeployed nurses' resilience in acute hospitals in Ireland, during a period of adversity (pandemic); to explore it's impact on their personal and professional identity, and their perception of meaningful supports and coping.

### Design

This is a qualitative IPA study exploring the thoughts, feelings, and perceptions of the nurse participants. The researcher drew upon IPA to explore a more existentially informed study. The two key principles of phenomenological psychology fit well with this research study as it focuses on intentionality, or 'what' is being experienced, and how one's biases add further constructs of meaning to the experience. Hermeneutics is the art of interpretation, and the second theoretical foundation of IPA where the researcher attempts to step into the shoes of the participants, though that may never be completely possible [36]. The third theoretical foundation, idiography, is the in-depth analysis of individual participants, and the examination of their perspectives in their own unique contexts. The researcher actively engaged with the transcripts in analysing what the participants were saying, but also at how they were saying it, allowing the researcher to read between the lines thus developing a thorough understanding of the personal experiences of each participant [36].

### Study participants

Six (n = 6) female nurses between the ages of 25 and 65 took part in this study, the author's master's dissertation. All participants were redeployed from their existing roles to COVID wards in acute Irish hospitals, during the pandemic, from two provinces in Ireland. The nurses were recruited, under very difficult clinical circumstances, via convenience and snowball strategies, which represented a homogeneous sample of participants in keeping with the theoretical pillars of IPA [37]. When it comes to IPA, Reid and colleagues suggests that 'less is more' in its commitment to ideography, and that the examination of fewer participants in greater depth is far more valuable than the simple descriptive analysis commonly seen in grounded theory, thematic analysis, or indeed in weak IPA studies [38]. Smith and colleagues report that sample size depends on study context and that each study should be considered on a case-by-case basis. They suggest that between three and six participants for master's level is sufficient [39].

Only female nurses who were redeployed to COVID wards in acute Irish hospitals were included in the study. The rationale for this was that female nurses who were redeployed to COVID wards, and who had significant caring

commitments at home, were at greater risk of psychological distress and harm [40]. Consequently, male nurses and student nurses were excluded from the study. Nurses who were not redeployed or who did not work on COVID wards were also excluded.

## Data collection

Recruitment for the study occurred over five weeks commencing on 28/02/2021 and ending on 03/04/2021 during the third wave of the pandemic. Written communication took place via the researcher's university email credentials and verbal correspondence took place via the researchers Microsoft Teams account. Following initial email contact, a short Microsoft Teams call took place to provide a brief outline of the study, obtain verbal agreement, form initial rapport, and address any queries. Participants also received information on the purpose and nature of the study, the aims and methodology, and information about the researcher. Following the calls, each participant was emailed a copy of the study information sheet and consent form as well as a copy of the interview schedule. Signed consent forms were returned to the researcher by email and stored securely on OneDrive, the university's secure server. The interviews took place over seven weeks between 15/03/2021 and 01/05/2021, during the latter end of the third wave of the pandemic. Participants were briefed again at the start of each interview on the reasons for doing the research, and that the information they provided during the study would be kept confidential, and that they reserved the right to withdraw from the study. The first author carried out the virtual semi structured interviews, lasting between 60 and 90 minutes (mean length = 75), using Microsoft Teams, due to lockdown protocols, and to protect the nurses. The recordings were split using Panopto software, video recordings deleted, and audio recordings were uploaded to OneDrive for short term storage to facilitate transcribing. Interview notes were also stored securely on OneDrive. Once transcribing in Microsoft Word had concluded, transcripts were forwarded to UCC via edugate's authenticated Filesender services for long term storage on an encrypted hard drive, and all associated OneDrive files were subsequently deleted. Microsoft Excel was used at the analysis stage and for table formatting. The interview schedule followed a topic guide consisting of four main domains; experience of and adaptation to adversity; self & identity, supports, and checking in and debriefing. Semi-structured interviewing was chosen as it allowed a purposeful conversation; beginning with general questions, moving to more personal questions once rapport was established. It also highlighting the subjective experiences of the participants, while also taking into consideration the use of language, cultural norms, and social aspects of the data [41].

## Ethical considerations

Ethical approval (application MCP 812202005) was obtained on 25/02/2021, from the School of Applied Psychology Research Ethics Committee (APREC), at University College Cork (UCC). Participants were informed prior to the interviews, the reasons for doing the research, and that the information they provided during the study would be kept confidential, and that they reserved the right to withdraw from the study. They were informed that their names would be removed by being assigned numerical codes during the transcription and coding process, and by using pseudonyms during data analysis. The researcher also checked what supports participants had access to, and provided additional support where required, and given the stressful nature of the participants' work, the researcher checked in with participants at the start and end of each interview.

## Data analysis

Interviews were transcribed verbatim by the main researcher (SC), and analysis took place by drawing upon the seven steps of IPA [39], to explore a more existentially informed study. The aim was to generate, categorize and organise potential themes from the data by reading and re-reading transcripts and listening to the audio recordings. The researchers

stepped inside the world of redeployed nurses working on COVID wards, to try and understand how they experienced it, and to examine how they made sense of it, as meaningful insights were crucial to the understanding, perspectives, and common experiences of each nurse [39]. Any queries from the findings were directed to the participants for clarification. Transcripts were analysed individually and were annotated by colour-coding and note taking to try and identify potential codes and themes described as descriptive, linguistic or conceptual concepts. The next step required the researcher to identify and label the emerging themes from the transcripts which represented the nature, quality and meaning of the nurses' experiences, including themes such as 'intense fear' and 'potent use of metaphors.' Coding was cross checked for errors and omissionsby the second researcher (AT) who was also the author's dissertation supervisor. The final steps involved an attempt to introduce structure where the researcher listed themes, and formed clusters with shared meanings and references, and others that were characterized by hierarchical relationships with each another. The researcher moved back and forth between themes and original text where some changes to codes were made in this process. The final step involved producing a summary table with quotations.

### Role of the researcher and study rigour

One of the researchers (SC) was a senior frontline nurse working in a critical COVID-19 role in the acute hospital setting. Because at times, she felt emotionally involved, and understood what her colleagues were going through, she maintained a journal and checked in with her supervisor AT, as a strategy for maintaining reflexivity, keeping track of reasoning, judgment, bias, and emotional reactions [42]. It should be noted that SC was analysing the "familiar" rather than the "unknown" and an advantage here was that she had an inherent feeling of what the participants were going through, and at times could recognise, and have the coaching competence to refer participants for external supports where required. The researcher SC implemented certain strategies to ensure quality, credibility, and trustworthiness during data collection and analysis which included an experienced researcher AT providing oversight and feedback on the interview protocol, analysis process and themes. Regarding transferability, within the write-up, the researcher SC provided adequate information for readers to evaluate relevancy, whilst also evoking a sense of shared experience. Further details can be viewed in supporting information.

### Results

Between March and May 2021, the first author interviewed six (n = 6) experienced nurse participants between the ages of 25 and 65 years. Five (n = 5) were Irish nurses and one (n = 1) was from outside Ireland but was working in Ireland for most of her nursing career. One participant was approaching retirement. Three experiential themes illustrated the nurses journey from firstly needing to find ways to protect themselves from the sacrifices on the frontline, to then moving to fortify themselves as a collective, and finally through experiencing critical turning points and growth. A summary of superordinate, subordinate, sub-themes and quotations on nurses' experiences on the frontline during the COVID-19 pandemic are listed in S1 Table and are further illustrated in the thematic mapping (S1 Fig).

### Protection of sacrificial self

Nurses found a need to protect themselves emotionally and mentally from the intense psychological overwhelm they experienced, which was triggered by the sacrifices they made on the frontline.

The emotional intensity arose from the nurses coming face-to-face with an existential threat, and the moral distress they experienced by being deployed to an unknown territory. Their fear was potent, especially the fear of infection and a fear of death and dying, for themselves, but also for their loved ones and patients.

> Well, you walk into a ward, you might not have been in there for three hours, you don't know what you're going to get … there are bodies going out every single day. There are just all these people crowding in. It's like "Mash" … coming in on trollies, barely able to breathe, frightened, anxious. I mean, it was a war zone. (Una)

Although they were redeployed to the frontline, the nurses showed a strong willingness to turn up and serve every day. This stemmed from a moral duty where their values got called into question in situations where they felt dangerously ill-equipped to cope.

I wasn't fighting the fact that I needed to go in and do this… I knew it was the right thing to do, so there was no doubt in my mind even if I got COVID. (Una)

Concerns about the nursing care deficits, meant that some even volunteered extra time with their patients to address this:

I sat with quite a lot of patients … so they wouldn't be on their own, which I wouldn't have done before, but I made an effort … I'd stay on after shift… and just give them that little bit extra attention and care, just because they weren't going to get it, when, when they did die. (Emma)

Claire portrayed a compassionate picture of how devastating it was for her, not to be able to provide dignified care at end-of-life. She suffered extreme guilt and despair at being forced to pack her patients' bodies into cadaver bags in their hospital gowns, which deeply wounded her, as it went against her core values:

I don't want people to go in double black bags … I feel like the dignity of the person at the time of their death, or even after they died. I'm not a grocery material to go in a plastic bag … I don't want to pack anymore bodies; I don't want to do that. (Claire)

All the nurses felt betrayed and lonely where they were left to face high risk and ethical dilemmas. Three of the nurses, in more critical frontline roles, felt forced into choosing which patients to save over others, and this came with huge emotional cost. "I took on everything, and if someone was deteriorating, it was because I wasn't doing a good enough job or if I had missed something…I felt like it would have been my fault". (Emma)

The nurses identified ways of protecting themselves emotionally to help maintain their focus at work. Some put their own emotions on hold whereas others just about 'held it together,' and all thought they would fall apart if they began processing their feelings too soon:

I can't let myself fully process the past year, because I don't know if I'm gonna fall apart or not, … am I gonna hate my job and never want to go back…if we start kind of questioning ourselves, am I OK to do this? (Trish)

Another protective mechanism was that most of them purposely detached from their chaotic surroundings. Ruth recalled how vital it was for her to "get off the merry-go-round" to recharge before getting back on. Three of the nurses detached by mentally hunkering down to unshackle themselves from the enormous mental strain. Once hunkered down, they entered into a purposeful flow state, with a heightened focus on critical tasks. This induced flow-state was protective in nature as it helped them to focus on the present moment, shelve their distractions, which helped sustain their energy levels:

My mantra was "one day at a time, one task at a time" and the only way I could get through my day, and if I lifted my head up to think of all the things that I had to do and all the responsibilities I had, I would have panicked and not being able to do anything, so I had to just break it down. "I'm here, I have to do this, no one else is going to do this, I have to just go in and do my best, there is nobody else" So, you need to just concentrate and try and get through the simple tasks that you're doing and then you go on to the next task and... my main aim was not to kill anybody. (Una)

Another way they protected themselves was by the potent use of metaphors which helped them to make sense of the complex situations they found themselves in. Quite often they likened traumatic events to war where they endured "the

scars" of battle by "working in the field". Trish gives us a sense of how courageous and fearless she felt going into battle with "all guns blazing" flanked by her peers.

## The fortifying effect of us

As a result of feeling isolated, unsupported, and left out in an unrelenting climate for far too long, the nurses turned to each other and began to develop a strong solidarity or "us", which helped them to bolster each other and build self-esteem against a growing perception of "them and us".

Frontline nurses were left feeling unsupported by a public who once portrayed them as heroes but now blamed them for spreading infection in hospitals. They also felt more isolated as they were the ones left holding the burden of responsibility, while others flouted guidelines or were spreading hate speech and vaccine misinformation on social media. This frustrated and angered Claire who wanted others to witness what was really happening behind hospital doors:

> Am I allowed to say that I would love to go slap people who say there is no suchthing as COVID? … because they haven't seen what we have seen in that place,they were not there holding the hands of a person that was dying. (Claire)

As a result of feeling isolated, all the nurses, over time, lost trust, and confidence in how the public supported them, and what began to emerge was a growing divide between "us", the nurses, and "them", the general public and media. This was further compounded by feelings of abandonment at work, because they felt excluded from key decision-making, and were not being adequately supported. Trish described how the lack of presence of meaningful leadership and supports, led to feelings of extreme hurt and loneliness, "to even see a face …, would have meant a lot to us… one visit... to the unit … it really made us feel aware of where we are in the pecking order". (Trish)

The nurses felt even more isolated as they perceived their redeployment as unfair, and resented being called to the frontline over and over again by senior management, when compared with other nursing colleagues who were never once redeployed.

> I think a lot of us just felt like it was a kick in the teeth… we really just felt like anumber… you know, redeployment was part of the gig, so "what do they have tocomplain about?" so yeah, it was just disheartening. (Emma)

The workload disparity and constant presence on the front line, was emotionally draining, placing an even bigger wedge between "us" the nurses, and "them", other healthcare professionals.

> It's more difficult because we're with the patient all day, I mean the rest of themwill do their treatment and … go, the doctor will come in and do their treatmentand go, the dietitian will come in and do her treatment and go, you know? So,we were the ones that were there all the time, yes, the constancy of it. (Denise)

In addition, the nurses were unable to lean fully on family and friends for support because they felt that they couldn't possibly have understood what they were going through. Although they weathered the same storm, they were not all in the same boat as the nurses. Nevertheless, both Una and Emma at times, felt their partners were "key" to getting them through emotionally:

> My poor boyfriend then got it all … if I wasquiet or upset … he knew when to push and pull back with things, which was great. (Emma)

At a certain point, things got so gruelling for the nurses that they began to question why they ever chose nursing as a career. Denise, who felt disrespected and undervalued, said, "I don't love nursing as much as I used to, and I would like…to maybe leave and pursue something else … I don't feel that my service was valued". (Denise)

What began to emerge for the nurses was a sense of power in numbers and solidarity. This is how they supported and bolstered themselves in a harsh climate, and began to build strength and confidence, feeling stronger against "them", and stronger against the world:

But from the lessons learned, I would (pause)… look after myself and look aftermy colleagues, that's all I can do … because that was one thing that was poorlydone, nobody ever had come back to us even asking us how we felt. (Claire)

The sense of "us" gave hope at a time when commitment was needed. Claire described how the strong bonds she had developed with her colleagues served as a lifeline to her when she needed support most, "We kind of like poured our hearts out and we were able to… vent our feelings with each other, to share our emotions". (Claire)

This solidarity delivered a huge amount of renewed self-assurance which helped protect and re-establish their self-esteem. Furthermore, the nurses also began to feel the positive relational effects of bonding together, "you're just tighter with your colleagues because you had to get through this together". (Una)

At times, some of the frontline clinical nurse managers (CNM) also became a part of us:

He was just one of those grounding people that if you saw him being calm andcollected, you knew you could be as well… He knew our abilities, he knew whatwe could and couldn't do, he knew when to check in on us … he was a greatvoice for us. (Emma)

Another CNM provided a positive road map forward which helped build inspiration and enthusiasm within the nursing team.

The CNM came up with an idea called the positivity board … It brought smiles to our faces, and I remember one of them writing "thank God the numbers are coming down", or … "I did not have to put anyone into a black bag today". (Clare)

Tributes at work were a stark reminder of how far the nurses had come and how lucky they felt to come through it:

There … were pictures of us, socially distanced together; so those pictures went up on the positivity board, something to look back and say "OK, fine we have survived this (Claire)

All the nurses felt a conjoined sense of meaning and purpose with their nursing colleagues around the world which gave them immense strength:

I think there's some kind of comfort there as well to know that it's happened all around the world and … most nurses have gone through it all together, so there is some kind of unity yeah, and like solidarity. (Emma)

## Critical turning points and growth

Being away from the frontline allowed them to look back and accept their feelings of fear, anger and overwhelm:

So, anger was the first thing, like there was no reason to be angry with everyone, I was angry within myself … I wasn't able to get on top of doing my things, which I would have normally done on a day-to-day basis looking after my patients. (Claire)

Engaging in positive self-talk was another turning point. Positive self-talk enabled them to feel less of a victim, reaffirm their contributions and to feel more in control, "I just had to talk to myself and tell myself I could do it and just get on with it and stay calm and stay focused" (Una)

Feeling better able to cope was another turning point. Claire felt if there was another COVID surge, she would be willing and happy to do it all again, having earlier felt she could never go through it again, "I don't want another wave coming, but if it does come to it, I would be willingly and happy going there". (Claire)

Another critical turning point was the ability to visualise, to feel hopeful. For example, Denise, was reminded in her own drawings that all waves eventually diminish, and that the safety of her lifejacket would carry her through:

I had one particularly bad shift … I remember getting up the next day … and painting surfing waves … and then there's this tiny little figure in a life jacket down at the bottom and that's me. It ended up that it really helped you know, because … I'm OK, I will survive this so. (Denise)

Similarly, Una expressed deep gratitude towards a child who had drawn a picture of a nurse superhero, and this struck a chord with her, giving her strength to keep going, "Thank you, our heroes, and I saw it and I just burst out crying like you know, that child, he doesn't know that, but…it did make a massive difference" (Una)

Another critical turning point was being able to reconcile with self:

I had feelings that the care delivered to the patients was sub optimal definitely, some of the time, not all of the time, and I had to reconcile that with myself with 'I did my best, I did my very best'. (Ruth)

A renewed meaning and purpose in their work and a strong sense of team spirit and pride was another turning point, "I'm proud of myself for going in and having put my shoulder to the wheel and, you know, pushed with the rest of my colleagues … I didn't run away". (Una)

Most of the nurses now felt more confident and vocal. For example, Emma's expert knowledge made her feel "more confident and definitely more resilient", with a strong desire to become a future nurse leader:

Well as a nurse, I think I've found my voice: if there is a sign of deterioration oranything … I feel more confident in myself. Definitely more confident in myability and my experience and my knowledge. (Emma)

What was strong for all the nurses was the pride and gratitude they felt for the work they had done:

I might have not done everything, but I was there for someone, and I've helped out, I have touched someone's life. I was able to offer that service, and I feel so good about it myself, that I have chosen this profession. (Claire)

All of the nurses begun to take great comfort in ordinary things, such as returning to their families after feeling so isolated on the frontline. They also felt a deep sense of "appreciation" for just being alive and connected with nature:

The sea is very calming, it's just like meditation, it's full of life, it's so positive, … it's the simple things that I enjoy …, baking in the kitchen, family and friends, a warm fire in the evening, a home cooked meal, barbecues in the garden, dogs, animals, cats. (Una)

The interviews for this study fostered new meaning and purpose for the nurses and helped them to make sense of their journeys. This brought about a renewed faith in peoples' desire to help and listen to their stories, "You touched on every

aspect of it and it's great, like, I feel this itself is a debriefing session … just kind of talk about it and thank you, thank you for doing such a study". (Claire)

## Discussion

Three key experiential themes were found that represented various stages of a journey taken by the redeployed nurses: first was the need to protect themselves emotionally and mentally from the intense psychological overwhelm they had experienced, having sacrificed themselves on the frontline; the second theme arose due to the nurses feeling isolated, and left out in a harsh pandemic climate for so long that they turned to each other for solidarity to bolster each other, and build self-esteem. What began to emerge was a sense of power in numbers which made the nurses stronger with each other and against the world. Finally, all of the nurses began to experience some critical turning points and growth.

### Protection of sacrificial self

This study gives insight into how redeployed nurses protected their emotional and mental wellbeing, in order to compensate for the sacrificial commitment they had to make, and the moral injury they experienced. Gaining insight into how the nurses protected themselves is important given that most of the literature has focused mainly on the sacrifices they made. In the current study, the redeployed nurses protected themselves from sacrificial commitment which was also found in some other studies that highlighted the nature and types of sacrifices made. For example, some studies have found that nurses experienced fear when at risk of contracting disease, thus impacting upon their emotional, physical and mental wellbeing as well as their capacity to do their job [11]. Whereas other studies found that the threat of death and dying creates a damaging heightened sense of alertness, and hypervigilance [12]. Other studies have reported moral injury, and exposure due to upholding a strong moral, professional, and spiritual work ethic [9]. In addition, the current study provides further insight into what nurse sacrificial commitment means; the sacrificial commitment mostly stemmed from the nurses' strong willingness to turn up every day, despite the overwhelming challenges and mandatory redeployment.

One of the key strengths in the current study is understanding how nurses protected themselves to counteract the sacrificial commitment they had to make on the frontline. This involved hunkering down, shielding or emotionally detaching, and making use of metaphors to portray both positive and negative circumstances. Regarding hunkering down, the nurses intentionally placed themselves in what could be described as a flow-type state. In other words, they entered an optimal psychological state that could be achieved to alleviate pressure, not just necessarily for enjoyment, which Csikszentmihalyi describes as 'a state in which people are so involved in an activity that nothing else seems to matter; the experience is so enjoyable that people will continue to do it even at great cost, for the sheer sake of doing it.' [43]. The current study also gives insight about the purpose of the hunkering flow state. This allows them to focus on the present moment thus distracting themselves from their surrounding chaos, allowing them to 'hold it together' and put their emotions on hold, so that they could continue to focus on the tasks at hand. Only one other study described a similar experience where ICU nurses were "pushing through long temporal episodes of work described as being in a "bubble" [44].

Protection of self was strong with the nurses in the current study who shielded themselves from risk in various ways. For example, they engaged in positive self-talk and mindful mantras during work to help sustain and maintain their hunkering down flow-like state. This fits with some literature which found that positive self-talk can increase work efficiencies whilst also reducing stress levels [45]. Also, the nurses in the current study shielded themselves from risk by emotionally detaching from work to recharge, getting enough rest, and staying in touch with family and friends. They also reduced their interactions with social media and news reports which is a common theme found in the wider nursing literature [15,25].

Another interesting way the nurses protected themselves emotionally was by using metaphors in an effort to make sense of, and move forward from, the emotional trauma they experienced. The richness of the metaphorical language

used by the nurses in the current study, was also evident in other nursing literature [10,16,17]. In addition, the nurses in the current study also used metaphors not only to reflect negative emotions, but also to reflect positive emotions such as courageousness, fearlessness and pride.

### The fortifying effect of us

Another key experiential theme in this study was the fortifying effect they gained from being an "us" against "them"; them being the general public, politicians, and at times their own family, and senior managers. Nurses turned towards each other over time to look for back-up and reassurance when public opinion and support had waned. Nurses found themselves mentally distancing themselves from the burden of the hero label in order to protect themselves. The above findings are consistent with some studies [15,44], where the hero status placed additional pressure on nurses to work even harder.

Also, the nurses turned to each other, to find strength and support as they often felt alone and isolated from their own family members, who didn't understand the distressing nature of their experiences with death and suffering. Although the nurses felt they were weathering the same storm, they were not in the same boat. Not surprisingly this was broadly consistent with the wider literature, such as where nurses found it hard to relate their feelings and experiences to their families as they just didn't understand what they were going through [33].

Another strengthening of a sense of us came as they felt solidarity together as a group who were experiencing strong feelings of exploitation and abandonment at work. They felt excluded from important decisions, and most were redeployed without choice, irrespective of their qualifications, personal strengths, and experiences. Unfairness was also felt, because some nursing colleagues were never once called upon by management for redeployment. This created resentment and unnecessary strain on relationships between the nurses (us), and some of their colleagues (them), as well as between nurses and senior hospital managers. This fits with what Billings found among redeployed staff, including nurses, who held resentment against others who were perceived to be doing less work [25]. Furthermore, as a collective, some of the nurses started to question why they ever chose nursing as a career and some contemplated leaving the profession altogether.

What emerged from the above findings were feelings of isolation, aloneness and largely feeling unsupported in a challenging landscape. Over time, the nurses began to build alliances with each other ("us") at a time when they needed to bolster their self-esteem and build resilience against 'them' and the world. This helped re-establish the self-esteem for which nurses were accustomed in their roles, and it gave them a renewed sense of hope at a time when huge commitment was required from them. This solidarity fits well with another study [44] which reiterates the importance of supporting one another to help cope with the situation.

At times, other staff felt part of this collective 'us', such as the frontline clinical nurse managers (CNMs), who sometimes brought further support. This helped build some inspiration and enthusiasm on the ground and can be seen in the wider literature where stronger collaboration on the ground helped improve nurse wellbeing [44]. This fits with the findings in one thematic study [46], where nurses experienced a stronger sense of 'them and us' with senior managers, where a lack of their presence and support was felt.

### Critical turning points and growth

The third experiential theme was that the nurses experienced critical turning points and growth as they wagered through the emotional turbulence of working during the pandemic. It has been strongly suggested that when nurses fail to do for their patients what they know deep down is good for them, their own moral integrity is at risk, and they end up with moral injury [47]. Not being able to alleviate pain and suffering or provide and care for critically ill patients during COVID has been a prevailing feeling among nurses in the current study, and moving beyond this phase was critical to their survival.

Key critical turning points for the nurses in this study included when they were able to step back and accept their feelings; enabling them to feel more hopeful or better able to cope, and being able to reconcile, or find purpose and meaning for themselves. This is widely consistent with older psychological constructs such as finding benefit [48] and enabling growth [15]. In the same way that grief and recovery do not follow an orderly process [49], the nurses began experiencing pivots and positive changes thus allowing them to regain some sense of self control in their lives. For example, Ruth recalled special moments with her patients in their last days before death and Una was deeply moved as she felt valued and a sense of pride after recalling children's drawing of nurses in capes. Both examples show the importance of emotional connectedness with nurse and patient. Empirical research has long identified emotional connectedness as a key coping resource for nurses [50].

Nurses in the current study conveyed a strong sense of duty toward their patients arising from a deep sense of needing to care [16,17]. This permitted them to feel immensely proud of themselves, and gave them a sense of purpose and meaning in their lives [4]. This was echoed widely in the literature where nurses were reported to having felt strengthened through adversity thus impacting positively upon patients' lives [51]. Moreover, Claire was able to rekindle her strong passion for nursing having lost it during her time on the front line. Harmonious passion is known to contribute to sustained psychological well-being, thus preventing negative affect [52], and the strong positive effect of believing in a higher power in times of crisis [28], may have contributed to Claire's growth. Some of the nurses said they now felt more expert in their roles. They felt listened to and respected, particularly by their medical colleagues, and this meant a lot as they felt more empowered to voice their opinions. This is consistent across the wider literature where the new roles nurses assumed during the pandemic provided significant reward [4].

The meaning of life is a fundamental factor of human existence, relating to the enormous existential power a person has to face in everyday challenges [53]. Denise expressed herself through art, Una and Emma made a grounding connection with nature and the outdoors. Others simply experienced a deep sense of gratitude and appreciation for getting through this and for life itself.

In terms of future research, it will be important in the coming years to track frontline nurses over time to see how they become resilient and overcome negative psychological effects or burnout. Future research could explore nurses spirituality to see if it extends their sacrificial duty, and to see if this better enables them to cope. In terms of positive organisational scholarship, it would also be interesting to know more about how nurses can purposefully enter a flow state or excelling zone, to enable them to detach and focus on the present moment to enhance their role. It may also be important to explore if future crises would merit redeploying nurses to areas that align with their core strengths and professional experience before being redeployed. Finally, given how cleverly nurses used time, space, and the extensive use of metaphors, a study examining the benefits of metaphor coaching for frontline nurses may be useful.

## Strengths and weaknesses

It could be argued that there are similar studies in the literature [54–57], however, those studies used more conventional approaches such as content analysis and thematic analysis, producing more generalised findings, and not interpretative phenomenological analysis, which involves a much closer examination of the experiences and meaning of the subject matter.

Moreover, one of the main strengths of this study was its capacity to give voice to frontline nurses who worked in Irish hospitals during the pandemic, whereas most of the studies outlined above took place in other countries. At the time of writing, this study was the first to examine the thoughts, feelings, perceptions and resilience of nurses working on the frontline in Ireland, through an interpretative phenomenological lense. Another key strength is that the researchers have suggested a more streamlined definition for 'nurse sacrificial commitment'. One of the most powerful findings was how the nurses purposefully entering into a flow-type state or excelling zone to detach from the surrounding chaos, which may enhance our knowledge on the psychological concept of "flow" in nurses during crisis.

One limitation with this study was that findings were drawn from six participants (n = 6) living and working in two provinces in Ireland. Although this offers a rich perspective from an Irish context, the researcher would have valued a wider scope of study participants but was unable to do so due to the overwhelming working conditions and time constraints on the nurses during the main COVID surge in 2021.

Another limitation was that male nurses working on COVID wards were not included in this study. Future research might include research on male nurses, or indeed a study comparing the effects of the pandemic on both female and male nurses. Some would argue that collecting qualitative data online can lead to limitations, however this was not our experience. In fact, participants reported that the online interviews were far more favourable than in-person interviews given the pandemic circumstances, and they felt much more at ease and protected from potential infection. All of the nurses valued deeply the online interviews as they felt they were the only meaningful form of support and debriefing they had received.

This research will help inform senior hospital managers and policy makers on how it was for nurses to work through a modern pandemic. One recommendation is that at-risk nurses would be provided with psychological support during the crisis and debriefing after the pandemic. If this is done well, it will present an opportunity for nurses to experience post-traumatic growth. In terms of recommendations for debriefing, it is the period directly after the crisis that is likely to be the most critical in terms of reducing the degree of psychological distress on nurses [30,31]. Other simple measures for managers to put in place include personally thanking staff, acknowledging their contribution, providing information on what help is available, and timely access to evidence-based care. It is important to move beyond just policy statements and put into practice the supports that are needed on the frontline.

Furthermore, another recommendation is the provision of more organisational planning by senior managers and policy makers, on how to effectively redeploy nurses to areas that align more with their core strengths, qualifications, and experience, before being redeployed. This recommendation is supported in the wider literature where strong collaboration from multidisciplinary teams in building organisational resilience and optimising redeployment methods is essential to nurse wellbeing [58]. This is also backed by evidence which suggests that patients would have received more specialised care, and nurses would have suffered less moral injury, anxiety and burnout, had nurse strengths and experience been considered before redeployment [59].

In terms of future studies, it will be important for researchers to carry out more longitudinal studies which will track the impact of poor emotional wellbeing and burnout in nurses over time. Also, other qualitative research could explore if nurses' sacrificial duty gets extended in a crisis. Other studies could examine if there is a correlation between nurse belief systems and sacrificial duty. It might also be useful to examine causal factors that facilitate nurse resilience and personal growth during sustained traumatic events.

There is little research dedicated to nurses and flow state, and how this enables them to focus on the present moment. We have seen already [60], where the relationship between the experience of flow at work and indicators of satisfaction, engagement and psychological well-being in nurses, have been beneficial. Further qualitative research is required on how nurses purposefully enter a flow state, or excelling zone, and how they manage to maintain that flow state.

Further research could compare if perceptions of senior managers are aligned with those of frontline nurses on aspects of their experiences on the frontline and how to effectively support each other.

Finally, given how nurses proficiently used time, space, and distance to dissociate themselves, as well as the extensive use of metaphorical narratives, a qualitative study examining the benefits of metaphor coaching for frontline nurses would be worthwhile.

## Conclusion

This study has given us a deeper insight into the journeys of each of the redeployed nurses, on how they protected themselves, given their enormous sacrifice on the frontline in Ireland during the pandemic. In facing that adversity, having been

left out in a harsh climate for so long, they assembled a collective sense of us against them, to protect and strengthen their resilience. In doing so, they also built self-belief, and supported one another, resulting in critical turning points and growth over time.

The study also exhibits how strongly the nurses engaged in various self-coping strategies, most notably by the extraordinary use of metaphors to "grasp the meaning of highly burdensome experiences", but also, by adapting daily work practices by purposefully entering into a hunkering down flow-like state or excelling zone to detach from the surrounding chaos.

Provision of supports for redeployed nurses, and indeed all healthcare workers, must be ongoing both during and after crisis events. Although there was strong evidence of peer-to-peer support, and support provided by a small number of frontline clinical nurse managers, none of the nurses in the current study had received any meaningful psychological supports from their organisations either during or in the aftermath of the pandemic, despite their sacrifice. Five years on, nurses in Ireland are still experiencing deeply, the negative mental health and wellbeing impacts of the COVID-19 pandemic. We must now direct our efforts towards how nursing professionals will be able to care for their patients, as well as caring for themselves as we face the next crisis which is looming on the horizon.

## Supporting information

**S1 Table. Summary table superordinate, subordinate and sub-themes & quotations.**
(DOCX)

**S1 Fig. Thematic map illustrating three major themes and subtheme.**
(TIF)

## Acknowledgments

To my six participants, you gave up so much to care for your patients and you became their family in their time of need. Thank you sincerely for your unrelenting sacrifice, and for your part in this research paper. Your stories have now been heard. Thank you also to my co-author and supervisor, Dr Anna Trace, for her support and guidance throughout the entire project.

## Author contributions

**Conceptualization:** Sinéad Creedon.

**Data curation:** Sinéad Creedon.

**Formal analysis:** Sinéad Creedon.

**Investigation:** Sinéad Creedon.

**Methodology:** Sinéad Creedon.

**Project administration:** Sinéad Creedon.

**Software:** Sinéad Creedon.

**Supervision:** Anna Trace.

**Validation:** Sinéad Creedon, Anna Trace.

**Visualization:** Sinéad Creedon.

**Writing – original draft:** Sinéad Creedon.

**Writing – review & editing:** Sinéad Creedon, Anna Trace.

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
