## [Decision Letter · Decision Letter 0]

24 Apr 2025

Dear Dr. Creedon,

We look forward to receiving your revised manuscript.

Kind regards,

Moustaq Karim Khan Rony, RN, MSS, MPH

Academic Editor

PLOS ONE

Journal Requirements:

4. We notice that your supplementary figure is uploaded with the file type 'Figure'. Please amend the file type to 'Supporting Information'. Please ensure that each Supporting Information file has a legend listed in the manuscript after the references list.

5. We note that there is identifying data in the Supporting Information file < S1 Table Participant characteristics.docx>. Due to the inclusion of these potentially identifying data, we have removed this file from your file inventory. Prior to sharing human research participant data, authors should consult with an ethics committee to ensure data are shared in accordance with participant consent and all applicable local laws.

-Location data

Please remove or anonymize all personal information, ensure that the data shared are in accordance with participant consent, and re-upload a fully anonymized data set. Please note that spreadsheet columns with personal information must be removed and not hidden as all hidden columns will appear in the published file.

6. We note that this data set consists of interview transcripts. Can you please confirm that all participants gave consent for interview transcript to be published?

If they DID provide consent for these transcripts to be published, please also confirm that the transcripts do not contain any potentially identifying information (or let us know if the participants consented to having their personal details published and made publicly available). We consider the following details to be identifying information:

- Names, nicknames, and initials

- Age more specific than round numbers

- GPS coordinates, physical addresses, IP addresses, email addresses

- Information in small sample sizes (e.g. 40 students from X class in X year at X university)

- Specific dates (e.g. visit dates, interview dates)

- ID numbers

Or, if the participants DID NOT provide consent for these transcripts to be published:

- Provide a de-identified version of the data or excerpts of interview responses

- Provide information regarding how these transcripts can be accessed by researchers who meet the criteria for access to confidential data, including:

a) the grounds for restriction

b) the name of the ethics committee, Institutional Review Board, or third-party organization that is imposing sharing restrictions on the data

c) a non-author, institutional point of contact that is able to field data access queries, in the interest of maintaining long-term data accessibility.

d) Any relevant data set names, URLs, DOIs, etc. that an independent researcher would need in order to request your minimal data set.

For further information on sharing data that contains sensitive participant information, please see: https://journals.plos.org/plosone/s/data-availability#loc-human-research-participant-data-and-other-sensitive-data

If there are ethical, legal, or third-party restrictions upon your dataset, you must provide all of the following details (https://journals.plos.org/plosone/s/data-availability#loc-acceptable-data-access-restrictions):

1. A complete description of the dataset

2. The nature of the restrictions upon the data (ethical, legal, or owned by a third party) and the reasoning behind them

3. The full name of the body imposing the restrictions upon your dataset (ethics committee, institution, data access committee, etc)

4. If the data are owned by a third party, confirmation of whether the authors received any special privileges in accessing the data that other researchers would not have

5. Direct, non-author contact information (preferably email) for the body imposing the restrictions upon the data, to which data access requests can be sent

**Additional Editor Comments:**

The reviewers suggested several revisions to improve the manuscript. Reviewer 1 recommended emphasizing the redeployment context more clearly in the title, abstract, and aim, and suggested that the introduction, while thorough, should be shortened and better organized. Minor wording and formatting issues were also noted, including the logical flow of certain paragraphs (e.g., Lines 112–116), the use of conjunctions, the ordering of references, and inconsistencies in reporting participant characteristics. Reviewer 2 emphasized the need to rewrite the introduction to clearly present the study's purpose, define key variables, and ensure coherence. Additional feedback included clarifying data collection details (e.g., interview timing), improving comparisons with existing literature—especially regarding themes like self-sacrifice and resilience—avoiding citations in the conclusion, simplifying the presentation of key findings, and including actionable recommendations for managers, policymakers, and future research. We kindly ask that you revise the manuscript accordingly, ensuring that all points are addressed to enhance clarity, structure, and scholarly contribution. Please see the detailed comments below.

Reviewers' comments:

Reviewer's Responses to Questions

**Comments to the Author**

1. Is the manuscript technically sound, and do the data support the conclusions?

Reviewer #1: Yes

Reviewer #2: Partly

2. Has the statistical analysis been performed appropriately and rigorously?

Reviewer #1: N/A

Reviewer #2: Yes

3. Have the authors made all data underlying the findings in their manuscript fully available?

Reviewer #1: Yes

Reviewer #2: Yes

4. Is the manuscript presented in an intelligible fashion and written in standard English?

Reviewer #1: Yes

Reviewer #2: Yes

Reviewer #1: Thank you for the opportunity to review this paper. It was thorough and well-written overall.

There is some new information in that the focus was nurses who were redeployed and it is set in Ireland. I think the fact that the participants were all redeployed should have been emphasized in the title, abstract and Aim. The introduction was very thorough but also very long.

Some specific suggestions include:

Line 69: "most exposed" Are you referring to most exposed to changes?

Line 83/84: I wouldn't assume that the nurses who were not redeployed "got to carry [on] as normal"

Line 101: "and" instead of &

Some paragraphs would benefit from a more logical flow/organization. One example is lines 112-116. The two sentences are very different and it isn't clear why they are together.

The references also seem out of order (18, 20, 21, 19, 22).

Line 133. Is this meant to be two sentences (ie. a period instead of a comma)?

Participant Characteristics: There is a high proportion of clinical nurse managers/educators. Is this typical for the nurses who were deployed?

Line 235 refers to ages 21-65 but this doesn't match the ranges reported in the Table.

The methods were described and the data transparency/analysis is sufficient and reader-friendly.

The quotes and presentation of the results were engaging and easy to follow.

Reviewer #2: Thank you for the opportunity to review this manuscript.

Introduction

-The introduction needs a major rewrite. I didn't understand your work. Please explain the need for your work.

-There is no need to state the introduction in separate items.Please state it coherently.Also define the main variables of the study.Summarize the introduction. It's too long.

Data Collection Gaps:

Unclear interview duration and timeframe (e.g., pandemic wave timing).

The findings of your study do not compare well with other studies. Please compare the discussion of self-sacrifice and resilience with your study.

-There is no need to cite the source in the conclusion section.

-In the conclusion, please state the main results in simple sentences.

Please also provide suggestions for managers, policies, and future studies.

**Do you want your identity to be public for this peer review?** For information about this choice, including consent withdrawal, please see our Privacy Policy

Reviewer #1: No

Reviewer #2: No

---

## [Author Response · Author response to Decision Letter 1]

7 Jun 2025

07/06/2025

Dear Editors,

Sincere thanks for your feedback on my manuscript. I was happy to revise the major changes, and I have included the following with this re-submission:

• A marked-up copy of my manuscript entitled 'Revised Manuscript with Track Changes' highlighting the changes made to the original version.

• An unmarked version of my revised paper without tracked changes. You should upload this as a separate file labelled 'Manuscript'.

• A pdf copy of this response to editors and reviewers.

In relation to the manuscript edits, please see my responses below:

1. I have reviewed my manuscript again to meet PLOS ONE's style requirements, including those for file naming. I hope that everything is now in order but if I have missed something specifically, please let me know.

2. We have indicated that there are indeed restrictions to data sharing for this study. To that end, we respectfully request to make our data sets available upon request.

a. De-identified extracts from qualitative transcripts are available only upon request as the participants consented to “extracts” from their interviews being quoted in the thesis, and in any subsequent publications, and not the full transcripts. Raw format (verbatim transcripts) contains both potentially identifying and sensitive participant information (e.g., detailed discussions, mental health challenges, potentially identifiable patient details, and participant geographical locations). Hence, there is a risk that sharing publicly would compromise participant anonymity and confidentiality. Data requests should be fielded to the School of Applied Psychology Research Ethics Committee (APREC), at University College Cork, at ethics.ap@ucc.ie.

The Data Availability statement has been updated accordingly in the submission form.

3. Please accept my apologies, I indicated that I would make my data available on acceptance; this was an error and oversight on my part. This is my first publication (apart from a review protocol) as a novice researcher and I am still navigating terminology and etiquette in relation to open access and data sharing. As per details outlined in section 2 and subsection a above, I now understand fully that PLOS One policy applies to all data except where public sharing would breach compliance with the protocol approved by my research ethics board. I have revised my statement to explain my reasoning and hope that the editor's input on an exemption will be permitted, to allow de-identified extracts from the qualitative transcripts be made available only on request.

4. Thank you for this. I have now updated my supplementary figure upload from file type 'Figure' to file type 'Supporting Information'. I have also re-checked that each Supporting Information file has a legend listed in the manuscript after the references list.

5. I have noted and agree with the deletion of Supporting Information file < S1 Table Participant characteristics.docx>.

6. As per section 2a above, participants only consented to “extracts” from their interviews being quoted in a thesis, and in any subsequent publications. We will therefore make our data sets available upon request. We will provide a de-identified version of the data or excerpts of interview responses, and we will also provide information regarding how this data can be accessed.

a) the grounds for restriction - Participants only consented to “extracts” from their interviews being quoted in a thesis, and in any subsequent publications. Furthermore, there is highly sensitive information on both participants and patients, in the participant transcripts.

b) the name of the ethics committee, Institutional Review Board, or third-party organization that is imposing sharing restrictions on the data - Data requests should be fielded to the School of Applied Psychology Research Ethics Committee (APREC), at University College Cork, at ethics.ap@ucc.ie.

d) Any relevant data set names, URLs, DOIs, etc. that an independent researcher would need in order to request your minimal data set – the doi link is also provided.

Response to Additional Editor Comments:

Thank you for highlighting your overall recommended revisions to improve our manuscript. We have responded to each point below.

Reviewer 1:

We have emphasizing the redeployment context more clearly in the title, abstract, and aim.

We have also shortened the introduction significantly and better organised the content and points.

We have reviewed the manuscript for minor wording and formatting issues, including the use of conjunctions and improved upon the flow of paragraphs. The order of references had to be edited significantly due to re-writing most of the introduction to reduce word count and make it easier to read.

Participant table has been removed and I can confirm that the participant characteristics are correct.

Reviewer 2:

The introduction was completely re-written. In doing so, we more clearly presented the purpose of the study, and defined more clearly the key variables to ensure more coherence.

We have further clarified data collection details and made significant improvements to improve our comparisons with the existing literature. We did focus specifically on self-sacrifice and resilience.

The citation was removed from the conclusion and was an oversight on my part.

The presentation of key findings have been presented in a simpler format and hopefully this improves the readers experience. We have added in actionable recommendation for managers, policymakers and future researchers.

Author Responses to Reviewers:

Response to Reviewer #1:

Thank you sincerely for your detailed review of our paper.

There is some new information in that the focus was nurses who were redeployed and it is set in Ireland. I think the fact that the participants were all redeployed should have been emphasized in the title, abstract and Aim - We have emphasizing the redeployment context more clearly in the title, abstract, and aim, and hope that this enhances our study.

The introduction was very thorough but also very long - we have shortened the introduction significantly and better organised the content and points.

Some specific suggestions include:

Line 69: "most exposed" Are you referring to most exposed to changes? – further clarification has been provided here.

Line 83/84: I wouldn't assume that the nurses who were not redeployed "got to carry [on] as normal" – we have re-phrased this for further clarification and context.

Line 101: "and" instead of & - corrected thank you.

Some paragraphs would benefit from a more logical flow/organization. One example is lines 112-116. The two sentences are very different and it isn't clear why they are together. – further clarification has been given here and throughout the manuscript. We hope this has improved the flow significantly.

The references also seem out of order (18, 20, 21, 19, 22) –The references had to be completely re-ordered due to re-writing the introduction and we have both given them a final check before re-submission. (Your reference to 18, 20, 21, 19, 22 above may just have been that a historical reference was subsequently referenced more than once in the manuscript).

Line 133. Is this meant to be two sentences (ie. a period instead of a comma)? – this has now been corrected.

Participant Characteristics: There is a high proportion of clinical nurse managers/educators. Is this typical for the nurses who were deployed? – In fact it was quite typical to have clinical nurse specialists / managers redeployed so as not to deplete the headcount of staff nurses from the main wards. These nurse managers and nurse specialists (e.g., cardiac, renal, neurosurgical, diabetes nurse specialists etc.) were redeployed from their specialist areas to work on the frontline while their specialist areas were effectively shut down during lockdown. I hope this answers your query.

Line 235 refers to ages 21-65 but this doesn't match the ranges reported in the Table – this has now been corrected.

The methods were described and the data transparency/analysis is sufficient and reader-friendly – noted thank you.

The quotes and presentation of the results were engaging and easy to follow – noted thank you.

Response to Reviewer #2:

Thank you very much for taking the time to review our paper.

The introduction needs a major rewrite. I didn't understand your work. Please explain the need for your work - On reading the introduction again, taking into consideration your recommendations, we both agree that the introduction needed to be re-written and shortened.

There is no need to state the introduction in separate items. Please state it coherently. Also define the main variables of the study. Summarize the introduction. It's too long. – we have removed the separate introduction sub-headings and stated the main points more clearly and coherently. We have re-defined the main variables and summarised the introduction by re-writing and shortening it.

Data Collection Gaps: Unclear interview duration and timeframe (e.g., pandemic wave timing). – we have now provided dates for data collection and included timeframes for interview duration, and also included which wave applied, to give further context.

The findings of your study do not compare well with other studies. Please compare the discussion of self-sacrifice and resilience with your study – we have made significant enhancements to improve our comparisons with the existing literature. We did focus specifically on self-sacrifice and resilience.

There is no need to cite the source in the conclusion section – noted and corrected thank you.

In the conclusion, please state the main results in simple sentences – this has now been addressed by re-writing the conclusion section to read more clearly.

Please also provide suggestions for managers, policies, and future studies – this has also been added in, thank you.

I hope I have fulfilled all of the requirements for re-submission but in the event I have omitted something or not clarified anything fully, please let me know so that I can address expediently. I look forward to hearing from you.

With Best wishes,

Sinéad

---

## [Decision Letter · Decision Letter 1]

11 Jul 2025

From protection of sacrificial self to critical turning points and growth: Redeployed nurses’ experiences on the frontline during the COVID-19 pandemic.

PONE-D-24-52634R1

Dear Dr. Creedon,

We’re pleased to inform you that your manuscript has been judged scientifically suitable for publication and will be formally accepted for publication once it meets all outstanding technical requirements.

Kind regards,

Moustaq Karim Khan Rony, RN, MSS, MPH

Academic Editor

PLOS ONE

Additional Editor Comments (optional):

Reviewers' comments:

Reviewer's Responses to Questions

**Comments to the Author**

Reviewer #1: All comments have been addressed

2. Is the manuscript technically sound, and do the data support the conclusions?

Reviewer #1: (No Response)

3. Has the statistical analysis been performed appropriately and rigorously?

Reviewer #1: (No Response)

4. Have the authors made all data underlying the findings in their manuscript fully available?

Reviewer #1: (No Response)

5. Is the manuscript presented in an intelligible fashion and written in standard English?

Reviewer #1: (No Response)

Reviewer #1: Thank you for your thoughtful response to my initial feedback. The manuscript is much clearer and reader-friendly.

**Do you want your identity to be public for this peer review?** For information about this choice, including consent withdrawal, please see our Privacy Policy

Reviewer #1: No

---

## [Editor Report · Acceptance letter]

PONE-D-24-52634R1

PLOS ONE

Dear Dr. Creedon,

I'm pleased to inform you that your manuscript has been deemed suitable for publication in PLOS ONE. Congratulations! Your manuscript is now being handed over to our production team.

Kind regards,

on behalf of

Mr. Moustaq Karim Khan Rony

Academic Editor

PLOS ONE